# The Influence of Nursing Home, Ward, and Eldercare Workers on the Number of Resident Handlings Performed per Shift in Eldercare

**DOI:** 10.3390/ijerph182111040

**Published:** 2021-10-20

**Authors:** Stavros Kyriakidis, Matthew L. Stevens, Kristina Karstad, Karen Søgaard, Andreas Holtermann

**Affiliations:** 1National Research Centre for the Working Environment (NRCWE), Lersø Parkallé 105, 2100 Copenhagen, Denmark; stk@nfa.dk (S.K.); mws@nfa.dk (M.L.S.); kristinakarstad@gmail.com (K.K.); 2Department of Sports Science and Clinical Biomechanics, University of Southern Denmark, Campusvej 55, 5230 Odense, Denmark; ksogaard@health.sdu.dk; 3Department of Clinical Research, University of Southern Denmark, Winsløwparken 19, 5000 Odense, Denmark

**Keywords:** patient handling, nurses, multi-level, healthcare, day shift, evening shift, day-to-day variation

## Abstract

The purpose of our study was to investigate which organizational levels and factors determine the number of resident handlings in eldercare. We conducted a multi-level study, stratified on day and evening shifts, including information on four levels: nursing homes (n = 20), wards within nursing homes (day, n = 120; evening, n = 107), eldercare workers within wards (day, n = 619; evening, n = 382), and within eldercare workers (i.e., days within eldercare workers; day, n = 5572; evening, n = 2373). We evaluated the influence of each level on the number of resident handlings using variance components analysis and multivariate generalized linear mixed models. All four levels contributed to the total variance in resident handlings during day and evening shifts, with 13%/20% at “nursing homes”, 21%/33% at “wards within nursing homes”, 25%/31% at “elder-care workers within wards”, and 41%/16% “within eldercare workers”, respectively. The percentage of residents with a higher need for physical assistance, number of residents per shift, occupational position (only within day shifts), and working hours per week (only within day shifts) were significantly associated with the number of resident handlings performed per shift. Interventions aiming to modify number of resident handlings in eldercare ought to target all levels of the eldercare organization.

## 1. Introduction

Musculoskeletal disorders (MSDs) are highly prevalent among the working-age population [1,2,3] and impose significant costs for the individual, workplaces, and society [4,5]. Eldercare workers represent a highly vulnerable working group for MSDs, where annual prevalence of low back pain and neck/shoulder pain is between 51%–71% and 31%–52%, respectively [6,7]. Additionally, eldercare workers have high rates of sickness absence and early retirement, predominantly attributed to MSD [8,9].

Resident handling activities comprise one of the core tasks of eldercare work [10]. These tasks include lifting, repositioning, turning, help with support stockings, pushing and pulling residents in portable chairs, and kneeling. Previous studies conducted in healthcare settings have found that the number of resident handling activities per work shift is between 7.5 to 9.5 times for repositioning and between 2.8 and 20 times for transferring [11,12,13]. These resident handling tasks are documented to increase the workers’ risk of MSD [14,15] and sickness absence [8].

Individual physical and psychosocial factors are associated with the pathogenesis of MSD among eldercare workers [16,17,18,19,20,21]. Other factors are related to the policies and available ergonomic aids for resident handling activities at the workplace (i.e., the organization level), or the staff ratio and the number of residents with a high need for resident handling activities who are being handled within a ward (i.e., a ward level organizational factor) [22,23].

Previous studies examining resident handling activities have usually been conducted with a one-level exposure approach (i.e., they do not consider the influence of higher organizational levels such as the ward or nursing home) [15,24,25,26,27,28,29]. Thus, eldercare organizations have limited evidence about which nursing home levels and factors need to be targeted in order to reduce the number of resident handlings performed per shift. Taking into account the complexity of the simultaneous influence of the organizational and individual factors in a multi-level approach is likely to provide a more holistic understanding to reduce the number of resident handlings.

Furthermore, the vast majority of studies investigating resident handling activities are primarily based on self-reported information [11,12,13,15,25,26], which can lead to imprecise and biased results. Only a few studies have used direct observations as their core methodology [30,31] and with assessments either conducted for three 2-hour periods [32] or among a limited number of participants [30]. A more comprehensive quantification of resident handlings utilizing observational exposure assessment tools may increase the evidence of factors associated with number of resident handlings.

Additionally, the majority of the studies assessing resident handling tasks among healthcare personnel were performed during day shifts only [30,31], hence not accounting for the possible differences between shifts. Research in the eldercare setting has demonstrated that there are notable differences between day and evening shifts [33]. Furthermore, day-to-day variation of the number of resident handlings performed during day and evening shifts has not yet been explored. Thus, it is highly important to investigate which specific nursing home levels and factors influence the number of resident handlings during day and evening shifts. Furthermore, possible differences may exist between days in eldercare, making it important to study the day-to-day variation of factors within eldercare work.

To the best of our knowledge, no studies have explored the influence and extent of organizational levels and factors on the number of resident handlings in eldercare using a multi-level methodology and direct observations. The purpose of this study was to investigate which organizational levels and factors determine the number of resident handlings in eldercare and to evaluate the influence of each level of the nursing home on the number of resident handlings performed during day and evening shifts. 

## 2. Materials and Methods

This is a cross-sectional multi-level study utilizing baseline data from the Danish Observational Study of Eldercare work and musculoskeletal disorders (DOSES). A detailed description of the study design, data collection, and methodology has previously been published [34]. DOSES received ethical approval from the Danish Data Protection Agency and the Ethics Committee for the Capital Region of Denmark (H-4-2013-028). 

### 2.1. Study Population

In total, 83 nursing homes located in the administrative Region Zealand, in the Greater Copenhagen area in Denmark, were invited to participate in DOSES. Twenty nursing homes (18 municipal and 2 private), including 126 wards and 941 eligible eldercare workers, agreed to participate. Eligibility criteria for participation in DOSES were: aged between 18 and 65 years old, employed more than 15 hours per week, working on day, evening, or rotational shifts, and allocation of at least 25% of their working time on activities related to direct resident care. Individuals that were on long-term sickness absence, pregnant, not permanently employed, or not spending a minimum of 25% of their working time on tasks related to the direct care of the elderly were excluded from the study. 

In order to maintain as much data as possible and minimize selection bias, the only criterion for the eldercare workers to be included in the present study analysis was filling in the work schedules at baseline providing information on which residents they had been taking care of over a three-week period. Thus, the final study population for the day shifts consisted of 20 nursing homes, 120 wards, 619 eldercare workers, and 5572 registered days. For the evening shifts, the final study population consisted of 20 nursing homes, 107 wards, 382 eldercare workers, and 2373 registered days (Figure 1).

### 2.2. Data Collection

Baseline data collection at the 20 nursing homes were conducted from September 2013 to December 2014. Data from each participating nursing home were collected over a period of 1 to 2 weeks. Organizational factors were collected at the ward level with a web-based questionnaire to the team managers. Eldercare workers’ factors were collected at the eldercare worker level with a structured self-administered questionnaire. Finally, information about the number of resident handlings performed per shift were collected through real-time workplace observations by trained observers. We linked data between the different levels by identification numbers for nursing home, ward, eldercare worker and day (date), which was registered for each outcome “Number of resident handlings performed per shift”.

#### 2.2.1. Organizational Factors

All organizational factors were collected at ward level. At baseline, team managers of each ward (41 team managers administered the 120 wards in the day shifts and 40 team managers administered the 107 wards in the evening shifts) answered a web-based questionnaire. Information on the number of residents located at the ward and the number of staff normally present at the ward in a shift, were used to calculate staffing ratio (total number residents divided by eldercare workers working in a shift) for day and evening shifts separately. Furthermore, information on the residents’ need for physical assistance (light, moderate, comprehensive, or complete) were provided by the team managers on standardized lists. We calculated the percentage of residents with high needs for physical assistance (i.e., comprehensive or complete) for each ward. During meetings with upper-and team-managers, we collected information regarding the type (somatic, dementia, rehabilitation, or independent living) of each ward. The type of ward was dichotomized (somatic versus dementia/other).

#### 2.2.2. Eldercare Worker Factors

We collected information on age, sex, occupational position (i.e., social service helper, social service assistant or other), and working hours mainly through standardized lists provided by the management. If the information were missing at the lists, we collected it, as with information on seniority, through a structured questionnaire provided to the eldercare workers at their workplace. At the “within eldercare workers” level (i.e., days within eldercare workers), the number of residents each eldercare worker were responsible for providing care per shift were drawn from work schedules (i.e., allocation of the residents between eldercare workers) collected over a three-week period at baseline.

#### 2.2.3. Number of Resident Handlings Performed per Shift (Outcome)

In DOSES, more than 4700 real-time workplace observations were conducted at baseline during a 4-hour period for the day shifts and during a 5-hour period for the evening shifts. The observations contained information on all caring activities needed for each resident at the 20 nursing homes, including information on the number and types of resident handlings. Types of resident handling activities included lifting, repositioning, turning, help with support stockings, push and pull resident in portable chair, and kneeling. The observations were conducted by trained observers following a strict protocol [35]. The observations were performed using tablets with the software Noldus Observer XT pocket observer (Noldus, Wageningen, The Netherlands). The inter-rater reliability of the observation instrument was shown to be good with corresponding agreement coefficients ranging from −0.27 to 1.0 [35].

Observations of the resident handlings required for each respective resident were merged with the information from work schedules, stating which residents each eldercare worker were responsible of providing care to each shift. We calculated the total number of resident handlings per shift performed by each eldercare worker as the sum of needed handlings of the residents they were responsible for.

### 2.3. Data Analysis

We used variance components analysis (VCA) to calculate the proportion of variance in the number of resident handlings performed per shift explained by the levels: nursing homes, wards within the nursing homes, eldercare workers within the wards and within eldercare workers (i.e., day-to-day variation). Nursing home, ward, and eldercare worker, were entered into the model as hierarchical random effects. Table 1 and Figure 1 illustrate and describe the four levels in the VCA. The analysis was stratified for day and evening shifts.

The proportion of variance is presented as “percentage of variability” and is calculated from the Between Group Variance (BGV) for each level and the Total variance. The intra-class correlation coefficient (ICC) is a descriptive statistic that quantitatively describes the proportion of variance that is explained by a grouping factor in multi-level / hierarchical data [36]:(1)$$\mathrm{Percentage}\;\mathrm{of}\;\mathrm{variability}=\mathrm{ICC}*100=\frac{\mathrm{Between}\;\mathrm{Group}\;\mathrm{Variance}\;\left(\mathrm{BGV}\right)}{\mathrm{Total}\;\mathrm{variance}}*100$$

To identify factors associated with the number of resident handlings performed per shift for day and evening shifts within each level of the nursing homes, we used the following two-step procedure. First, univariate analyses were conducted with each potential determinant added individually as a fixed-effect to the developed VCA model (i.e., retaining nursing home, wards, eldercare worker, and within eldercare workers as hierarchical random-effects). All variables with a p-value less than 0.10 were selected for further investigation. Second, we constructed a multivariate model by forward selection starting with the variables with the lowest p-value. Variables with a p-value less than 0.10 and resulting in the lowest -2loglikelihood were retained in the final model. We set the significance level for variable selection at 0.10 so that critical variables relevant to the outcome and of practical and / or clinical relevance are not omitted.

The extent of missing data differed across each variable and to keep most data in the analysis we analyzed the organizational, eldercare worker, and within eldercare worker factors with a different sample size for every variable (i.e., age, day = 5381, evening = 2309; sex, day = 5417, evening = 2310; seniority, day = 4889, evening = 1860; working hours per week, day = 5326, evening = 2274). However, the extent of missing data was not substantial.

To identify the percentage reduction of different sources of variance to the total variability in the resident handlings per shift for day and evening shifts in eldercare work across all levels, the variables that were found statistically significant in the multivariate model were individually introduced to the raw VCA model. The percentage reduction across all levels was calculated as the difference of the estimate of the between group variance of the raw VCA model subtracted by the estimate of the between group variance of the raw VCA model including the individual statistically significant factors.

For descriptive statistics, SPSS (IBM SPSS Statistics for Windows, Version 24.0. Armonk, NY: IBM Corp.) was used. All other analyses were conducted using the procedure PROC MIXED in SAS version 9.4 software (SAS Institute, Cary, NC, USA).

## 3. Results

### 3.1. Descriptive of the Hierarchical Structure and the Clusters within the Four Levels of the Dataset

In total, 619 eldercare workers within 120 wards from day shifts and 382 eldercare workers within 107 wards from evening shifts filled in work schedules. The eldercare workers working in day shifts registered on average 9 days of information, giving a total of 5572 days of registration. Accordingly, the eldercare workers working in evening shifts registered on average 6 days of information, giving a total of 2373 days. As the analysis of this study is stratified on day and evening shifts, it is only possible for the eldercare workers to have one registration each day. This means that the number of days registered by the eldercare workers and the corresponding measured day-to-day variation is equivalent to the number of shifts registered by eldercare workers and shift-to-shift variation. We provide an overview of the study participants and number of measured days, their hierarchical structure, and clustering in Table 1.

### 3.2. Descriptive Characteristics of the Study Population

Within the 20 participating nursing homes, the majority of the wards were somatic (day shifts 75%; evening shifts 75%) and the rest were dementia/other (day shifts 25%; evening shifts 25%). The average (SD) staff ratio (residents per worker) was 3.4 (0.9) for the day shifts and 7.5 (2.6) for the evening shifts. The eldercare workers were generally middle aged (day shifts’ mean [SD] = 44.4 [10.8]; evening shifts’ mean [SD] = 47.3 [11.0]), predominately female (day shift 95%; evening shift 94%) and most were employed as social service helpers (day shifts 42%; evening shifts 47%). The average (SD) job seniority (in years) was 15.1 (11.1) for the day shifts and 16.1 (11.2) for the evening shifts. The average (SD) working hours per week was 33.1 (3.2) for the day shifts and 29.5 (3.4) for the evening shifts. Table 2 illustrates the characteristics of the organizational factors on ward level and eldercare workers stratified by day and evening shifts.

### 3.3. Explained Variance at the Nursing Home, Ward, Eldercare Worker and within Eldercare Worker Levels

All four levels within the nursing homes contributed to the total variance of the number of resident handlings performed per shift during day and evening shifts (see Table 3). We found clear differences in the percentage of variance between day and evening shifts. The greatest source of variance occurred within eldercare workers (day-to-day variation; 41.4%) for the day shifts and between wards within nursing homes (33.2%) for the evening shifts. The rest of the variance for the day shifts occurred, in descending order, between eldercare workers within wards (25.0%), between wards within nursing homes (20.9%), and between nursing homes (12.7%). Accordingly, the rest of the variance for the evening shifts occurred, in descending order, between eldercare workers within wards (31.3%), between nursing homes (19.9%), and within eldercare workers (day-to-day variation; 15.7%). 

### 3.4. Determinants of Resident Handlings Performed per Shift Stratified on Day and Evening Shifts

For the organizational factors, an increased percentage of residents with a higher need for physical assistance was associated with a higher number of residents handlings for both day (β = 0.07; 95% CI 0.04 ; 0.10; p = <0.0001) and evening (β = 0.09; 95% CI 0.05 ; 0.13; p = <0.0001) shifts. 

Regarding the eldercare workers factors, being a social service assistant was associated with lower resident handlings only for day shifts (β = −0.65; 95% CI −1.24 ; −0.06; p = <0.05), when compared with being a social service helper. Furthermore, a higher number of working hours per week was positively associated with an increased number of resident handlings for day shifts only (β = 0.10; 95% CI 0.01 ; 0.19; p = <0.05). 

Finally, for the within eldercare workers factors, the number of residents per shift was a significant predictor for the number of resident handlings for both day (β = 1.47; 95% CI 1.37 ; 1.57; p = <0.0001) and evening shifts (β = 0.82; 95% CI 0.76 ; 0.87; p = <0.0001). Table 4.

### 3.5. Percentage Change of Different Sources of Variance to the Total Variability in Resident Handlings per Shift

For the organizational factors, when the percentage of residents with higher need for physical assistance determinant was added in the crude VCA model, a greater percentage reduction in the total variance in resident handling per shift was observed between nursing homes for the day shifts (44%) and evening shifts (62%). For the eldercare worker factors, when the occupational position determinant was introduced to the crude VCA model, a percentage reduction in the total variance in resident handlings per shift was observed between nursing homes (6%), between wards (1%), and between eldercare workers (3%) for day shifts only. In contrast, increased total variance of resident handlings was observed between nursing homes (−3%), between wards (−7%), and between eldercare workers (−1%) for day shifts only, when the working hour per week determinant was added to the crude VCA model. Finally, for the within eldercare workers factors, the greater percentage of the total variance in resident handlings per shift occurred between the eldercare workers (15%) for the day shifts, and between nursing homes (43%) for the evening shifts. Table 5.

## 4. Discussion

### 4.1. Summary of Findings

This cross-sectional multi-level study identified that all levels of the nursing homes (i.e., nursing homes, wards within nursing homes, eldercare workers within wards, and within eldercare workers) contributed to the total variance in the number of resident handlings performed per shift for day and evening shifts. Furthermore, we found that an increased percentage of residents with a higher need for physical assistance (both day and evening shifts), being a social service helper (day shifts only), a higher number of working hours per week (day shifts only), and an increased number of residents per shift (both day and evening shifts) were significantly associated with a higher number of resident handlings performed per shift.

### 4.2. Variance in the Number of Resident Handling Performed per Shift at All Hierarchical Levels

To the best of our knowledge, this is the first comprehensive multi-level study to investigate to what extent the number of resident handlings performed per shift is attributed to four hierarchical levels of nursing homes (i.e., nursing home, wards within nursing homes, eldercare workers within wards, and within eldercare workers). No previous studies have demonstrated that all four investigated levels of nursing homes have contributed to the total variance in the number of resident handlings performed during day and evening shifts.

The main result is that during day shifts, the highest variance in the number of resident handlings was observed within and between eldercare workers. There are only few studies using multi-level methodology to investigate how different hierarchical levels such as nursing home, ward, and eldercare workers, and registered days may influence the number of resident handlings. However, in accordance with our results, a previous study by Koppelaar et al. (2012) found that across all types of resident handlings (i.e., transfers, personal care of patients, repositioning, putting on or taking off anti-embolism socks) the greatest source of mechanical load was observed within nurses’ factors during day shifts only [37]. The organizations and the wards within them had limited contribution to the total variance in the mechanical load across all resident handling activities. This finding suggests that within and between eldercare worker factors are the main drivers to the number of resident handlings performed in nursing homes.

Although nursing homes in Denmark are under the governance of the assigned municipality, they present a high degree of independence, which could explain some of the variance in the number of resident handlings found between nursing homes during both day and evening shifts. However, Danish nursing homes have to follow standardized protocols and eldercare workers have to comply with certain principles and guidelines, regulated by the municipalities. This could explain the low level of variance in the number of resident handlings contributed by the nursing homes.

On the other hand, for the evening shifts, we found that the greatest source of variance in the number of resident handlings was observed between wards within nursing homes and between eldercare workers within wards. The variance could be due to the decreased number of eldercare workers during evening shifts. The observed differences in the variance in the number of resident handlings performed per shift during day and evening shifts are important, as it suggests that the number of resident handlings performed during day and evening shifts are highly driven by the different hierarchical levels of nursing homes.

### 4.3. Determinants of the Number of Resident Handlings Performed per Shift

Several determinants across all hierarchical levels of nursing homes were significantly associated with the number of residents handlings performed per shift. An increased percentage of residents with high demands for physical assistance (day and evening shifts), a higher number of working hours per week (day shifts only), and a higher number of residents per shift (day and evening shifts) were positively associated with an increased number of resident handlings performed per shift. Having an occupational position as a social service assistant compared with a social service helper was associated with less resident handlings per shift (day shifts only). For some of the investigated factors, our results are in accordance with previous studies. Koppelaar et al. (2012) found that a higher ratio of nurses per patient was associated with a decreased frequency of resident handling activities, such as manual lifting of patients, personal care of patients, patient transfers, and putting on and taking off anti-embolism stockings [37]. However, in that study, the determinants were investigated only for day shifts, compared with our study, which included both day and evening shifts.

The finding that residents with lower physical functional level are associated with more resident handling activities reflects that the demands of the eldercare occupation are closely related to the need for care among the residents. Engaging the resident during resident handling activities will potentially promote their mobility and decrease the need for physical assistance, thus reducing the physical demands on the eldercare workers during a shift [38].

Our finding that a higher number of residents per shift is associated with an increased number of resident handlings is highly intuitive. Increasing the staff ratio during a shift will potentially decrease the number of resident handlings performed per shift. 

### 4.4. Strengths and Limitations

The major strengths of this study are the large sample size and the multi-level design, which provides important insights into understanding the levels that contribute to the number of resident handlings performed per shift in eldercare work. Data on organizational and eldercare worker factors were collected from independent sources on each level and not with the use of questionnaires solely to eldercare workers. Furthermore, data about the number of resident handlings per shift were collected through real-time observations minimizing recall and measurement bias. However, a major limitation is the cross-sectional design, which prohibits attribution of causality to any detected association.

### 4.5. Practical Implications

This study has identified multiple factors across different levels of nursing homes associated with the number of resident handlings in eldercare work. Some of these factors (i.e., percentage of residents with a higher need for physical assistance) are not easy to change due to their nature. However, targeting modifiable factors such as reducing the number of residents assigned to each eldercare worker per shift or allocating the residents better between the eldercare workers can potentially reduce the fraction of eldercare workers with very high number of resident handlings per shift. Our results may be used to prevent excessive workload, pain, and sickness absence related to a high number of resident handlings performed per shift among eldercare workers. 

## 5. Conclusions

In the present study, all levels of nursing homes (i.e., nursing homes, wards within nursing homes, eldercare workers within wards, and within eldercare workers) contributed to the total variance in resident handlings during day and evening shifts. Furthermore, a higher number of resident handlings was significantly associated with the percentage of residents with higher need for physical assistance, the number of residents per shift, the occupational position (only within day shifts), and working hours per week (only within day shifts). These findings can be leveraged in order to reduce the number of resident handlings per shift by increasing the staff ratio, optimal workforce planning, and resource allocation within wards and increasing the use of assistive devices.

## Figures and Tables

**Figure 1 ijerph-18-11040-f001:**
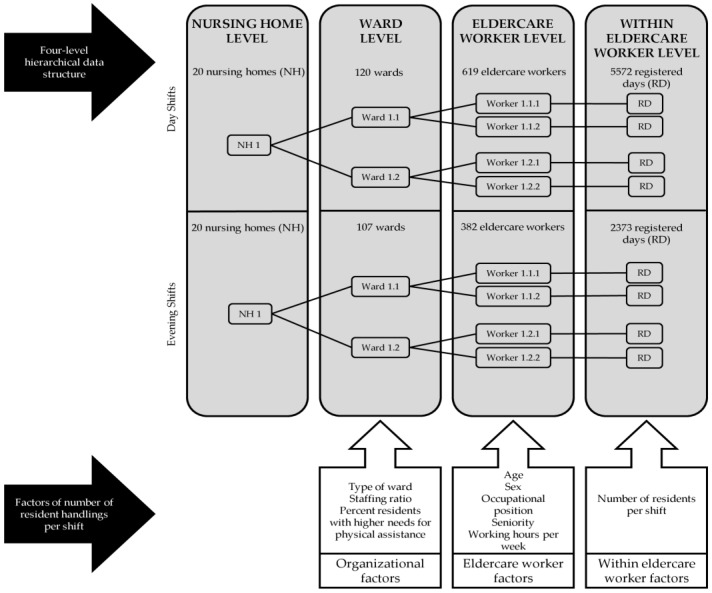
Scheme of the four-level hierarchical data structure of the present study and the level in which the specific investigated factors are measured at. Registered days (RD) (level 4) are clustered within eldercare workers (level 3), eldercare workers are clustered within wards (level 2), and wards are clustered within nursing homes (NH) (level 1), all stratified on day and evening shifts.

**Table 1 ijerph-18-11040-t001:** Descriptive of the hierarchical structure and the clusters within the four levels of the dataset. N = Total population.

Level	Day Shifts	Evening Shifts
**Nursing homes**	N = 20	N = 20
**Wards**	N = 120	N = 107
Wards within the nursing homes		
Mean	6	5.3
Median	5	4
Minimum	2	1
Maximum	12	11
**Eldercare workers**	N = 619	N = 382
Eldercare workers within the wards		
Mean	5.2	3.6
Median	5	3
Minimum	1	1
Maximum	14	11
**Within eldercare workers (days)**	N = 5572	N = 2373
Number of days within the eldercare workers		
Mean	9	6.2
Median	10	6.5
Minimum	1	1
Maximum	34	16

**Table 2 ijerph-18-11040-t002:** Characteristics of organizations on ward level and eldercare workers stratified on day and evening shifts work. (Res. = Respondents/cases; SD = Standard Deviation; % = percent; n = number.)

	Day Shifts			Evening Shifts		
	Res.	Mean (SD)	% (n)	Res.	Mean (SD)	% (n)
**Organizational factors** Type of ward Dementia/other Somatic Staffing ratio (residents per worker) Percentage of residents with higher needs for physical assistance **Eldercare worker factors** Age (years) Sex (female) Occupational position Social Service Helper Social Service Assistant Other Seniority (months) Working hours per week **Within eldercare worker factors (measured each shift)** Number of residents per shift Number of resident handlings per shift	120 120 120 584 588 619 516 578 5572 5572	3.4 (0.9) 41.0 (22.5) 44.4 (10.8) 181.5 (132.9) 33.1 (3.2) 3.5 (1.4) 6.6 (6.4)	25 (30) 75 (90) 96 (563) 42 (262) 39 (239) 19 (118)	107 107 107 361 362 382 291 357 2373 2373	7.5 (2.6) 39.3 (21.8) 47.3 (11.0) 193.3 (134.2) 29.5 (3.4) 7.5 (2.9) 8.7 (6.4)	25 (27) 75 (80) 94 (339) 47 (178) 29 (110) 25 (94)

**Table 3 ijerph-18-11040-t003:** Estimated contribution of different sources of variance to the total variability in number of resident handlings performed per shift in eldercare work stratified on day (N = 5572) and evening shifts (N = 2373) work. (BGV. = Between Group Variance; 95% CI = 95% confidence intervals.)

Sources of Variance	Day Shifts			Evening Shifts		
	BGV.	95% CI	Percentage of Variability	BGV.	95% CI	Percentage of Variability
Between nursing homes	5.4	2.6 ; 18.2	12.7	8.6	3.8 ; 34.4	19.9
Between wards within nursing homes	9.0	6.5 ; 13.3	20.9	14.3	9.8 ; 22.8	33.2
Between eldercare workers within wards	10.7	9.2 ; 12.6	25.0	13.5	11.4 ; 16.2	31.3
Within eldercare workers (Day-to-day variation)	17.8	17.1 ; 18.5	41.4	6.8	6.3 ; 7.2	15.7
Total variation	42.9	-	100.0	43.0	-	100.0

**Table 4 ijerph-18-11040-t004:** Univariate and multivariate analysis of the association between organizational, eldercare worker factors, within eldercare worker factors and the number of resident handlings performed per shift stratified on day and evening shift work. (Est. = Estimate; S.E. = Standard Error; 95% CI = 95% confidence intervals.)

	Univariate				Multivariate			
	Day Shifts (N = 5572)		Evening Shifts (N = 2373)		Day Shifts (N = 5326)		Evening Shifts (N = 2274)	
	Est. (S.E.)	95% CI	Est. (S.E.)	95% CI	Est. (S.E.)	95% CI	Est. (S.E.)	95% CI
**Organizational factors**								
Type of ward ^b^								
Dementia/other	REF		REF		REF		REF	
Somatic	1.89 (0.79) **	0.33 ; 3.45	1.97 (1.05) *	−0.12 ; 4.06	0.99 (0.77)	−0.54 ; 2.52	0.46 (0.95)	−1.43 ; 2.36
Staffing ratio (residents per worker) ^a, c^	1.09 (0.38) ***	0.34 ; 1.84	0.09 (0.19)	−0.29 ; 0.48	0.46 (0.38)	−0.29 ; 1.20	-	-
Percentage of residents with higher needs for physical assistance ^a^	0.06 (0.01) †	0.04 ; 0.09	0.10 (0.02) †	0.06 ; 0.14	0.07 (0.01)†	0.04 ; 0.10	0.09 (0.02)†	0.05 ; 0.13
**Eldercare worker factors**								
Age (years) ^a, d^	−0.02 (0.01)	−0.04 ; 0.01	0.01 (0.02)	−0.03 ; 0.04	-	-	-	-
Sex (female) ^b, e^	0.54 (0.69)	−0.82 ; 1.90	−0.28 (0.88)	−2.00 ; 1.44	-	-	-	-
Occupational position ^b^								
Social Service Helper	REF		REF		REF		REF	
Social Service Assistant	−1.07 (0.31) ‡	−1.69 ; −0.45	−0.91 (0.49) *	−1.87 ; 0.04	−0.65 (0.30) **	−1.24 ; −0.06	−0.35 (0.43)	−1.20 ; 0.50
- Other	−1.49 (0.41) ‡	−2.30 ; −0.68	0.40 (0.52)	−0.62 ; 1.43	−0.72 (0.44)	−1.59 ; 0.15	0.35 (0.49)	−0.62 ; 1.33
Seniority (months) ^a, f^	−0.00 (0.00)	−0.00 ; 0.00	0.00 (0.00)	−0.00 ; 0.01	-	-	-	-
Working hours per week ^a, g^	0.11 (0.05) **	0.01 ; 0.20	−0.23 (0.07) ***	−0.37 ; −0.09	0.10 (0.05) **	0.01 ; 0.19	−0.08 (0.06)	−0.20 ; 0.04
**Within eldercare worker factors (measured each shift)**								
Number of residents per shift ^a^	1.50 (0.05) †	1.40 ; 1.60	0.85 (0.03) †	0.79 ; 0.91	1.47 (0.05) †	1.37 ; 1.57	0.82 (0.03)†	0.76 ; 0.87

* p = <0.1; ** p = <0.05; *** p = <0.01; ‡ p = <0.001; † p = <0.0001; ^a^ = Continuous variable; ^b^ = Categorical variable; ^c^ Not included in the multivariate model for evening shift; ^d^ Number of cases without missing data, day = 5381 and evening = 2309 (only for the univariate model); ^e^ Number of cases without missing data, day = 5417 and evening = 2310 (only for the univariate model); ^f^ Number of cases without missing data, day = 4889 and evening = 1860 (only for the univariate model); ^g^ Number of cases without missing data, day = 5326 and evening = 2274 (only for the univariate model).

**Table 5 ijerph-18-11040-t005:** Percentage change to the total variability in resident handlings per shift (at each level), due to the inclusion of specific factors.

	Day Shifts (N = 5326)				Evening Shifts (N = 2274)			
	Nursing Home	Wards	Eldercare Workers	Within Eldercare Workers	Nursing Home	Wards	Eldercare Workers	Within Eldercare Workers
**Organizational factors**								
Percentage of residents with higher needs for physical assistance ^a^	44%	11%	0%	0%	62%	11%	1%	0%
**Eldercare worker factors**								
Occupational position ^b, c^	6%	1%	3%	0%	-	-	-	-
Working hours per week ^a, c^	−3%	−7%	9%	−1%	-	-	-	-
**Within eldercare worker factors (measured each shift)**								
Number of residents per shift ^a^	3%	−1%	9%	15%	43%	−2%	30%	26%

^a^ = Continuous variable; ^b^ = Categorical variable; ^c^ Not statistically significant in the multivariate model in evening shifts; Note: This table shows how the percentage variance derived from each level changes when adding individual specific variables into the crude VCA model. For example, adding the percentage of residents with higher needs for physical assistance variable into the crude VCA model reduced the total variance in resident handlings explained at the nursing home level by 44% for the day shifts.

## Data Availability

The study datasets are available at the Danish National Archives, https://www.sa.dk/en/k/about-us.

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
