# Peer review of "The Influence of Nursing Home, Ward, and Eldercare Workers on the Number of Resident Handlings Performed per Shift in Eldercare"

_ijerph, 2021, doi:10.3390/ijerph182111040_

Round 1

Reviewer 1 Report

Thank you very much for the opportunity to review this manuscript! Below are my recommendations:

  • Line 98: delete “possible” before “selection bias”
  • Line 195: I would suggest to provide the inter-rater reliability coefficient
  • Table 1: because you are reporting whole numbers, no need to put 0 after the decimal
  • Lines 602-603: unclear what is meant by “different hierarchical levels of nursing homes”
  • Lines 613-615: you already stated this earlier in the Discussion
  • Line 625: should be a comma before “thus,” not a period
  • Line 656: “contributed” – not “contributing”
  • Practical Implications and Conclusion: I would suggest to more precisely and succinctly summarizing what were the 1-2 core findings? Otherwise, the way it reads, it appears too diffuse and the reader cannot be left with a single most important takeaway message

Reviewer 2 Report

In this manuscript is presented the influence of nursing home, ward, and eldercare worker on the number of resident handlings performed per shift in eldercare. The topic is very relevant for eldercare and the study design and the study procedure are very clear. The article has a clear language and the aim of the study it is clear and interesting. I have, only, minor suggestions for revision:

Materials and methods

Line 212: “Percent of variability = ICC*100”. Please define ICC for the readers to understand the initials ICC

Line 218: “All variables with a p-value less than 0.10” Why the authors used p-value 0.10 and not 0.05? Normally, it is used the p-value less than 0.05. It remains to be explained why the authors choose this p-value.

Conclusions

Line 655: “However, the  levels ”eldercare workers within wards” and “wards within nursing homes” contributing the most to the variance in resident handlings for the day and evening shifts, respectively”. This sentence is strange, please rephrase its.

Although the authors referred practical implications of the work in the discussion, I suggest that they ground the conclusion with more concrete practical implications.
